# Dose–Response Relationship of Resistance Training on Metabolic Phenotypes, Body Composition and Lipid Profile in Menopausal Women

**DOI:** 10.3390/ijerph191610369

**Published:** 2022-08-20

**Authors:** Ana Carla Leocadio de Magalhães, Vilma Fernandes Carvalho, Sabrina Pereira da Cruz, Andrea Ramalho

**Affiliations:** 1Center of Micronutrients Researche, Josué de Castro Nutrition Institute, Federal University of Rio de Janeiro, Rio de Janeiro 20000, Brazil; 2Kinanthropometry Laboratory, Belo Horizonte Campus, Salgado de Oliveira University, Belo Horizonte 30170, Brazil

**Keywords:** resistance training, cholesterol, body composition, metabolically healthy obesity, menopause

## Abstract

The relationship between volume training of resistance training (RT), body composition and cardiometabolic profile in menopausal women is poorly understand. This study aimed to evaluate the dose–response relationship of RT on lipid profile, body composition and metabolic phenotypes in menopausal women. A total of 31 women were categorized according to different volume of RT. Body composition was evaluated by DEXA and the cardiometabolic risk by metabolic phenotypes and lipid profile. There was a higher frequency of metabolically unhealthy phenotype in women who practiced RT for less than two years and had a weekly frequency lower than three days a week (*p* > 0.05). Women with more than two years and a higher weekly frequency of RT had lower trunk fat mass than their counterparties (15.33 ± 7.56 versus 10.57 ± 4.87, *p* = 0.04; 16.31 ± 7.46 versus 10.98 ± 5.49, *p* = 0.03, respectively). There was an association between HDL-c and time of RT in years. A moderate correlation was identified between variables of body adiposity, time in years and weekly frequency of RT. The present study concludes that more time in years and weekly frequency of RT practice are associated with lower body adiposity in menopausal women, the first also being associated with HDL-c.

## 1. Introduction

Menopause is a period associated with intense physiological and body composition changes that favor the development of cardiometabolic complications among which are coronary artery disease, type 2 diabetes mellitus (DM2), systemic arterial hypertension (AH) and metabolically unhealthy phenotype (MUH) [1,2,3]. This metabolic risk profile is characterized by inadequacies in cardiometabolic and inflammatory parameters that are associated with morbidity and mortality. However, there are women of non-reproductive age who may be exempt from these alterations and are considered metabolically healthy (MH) [4,5].

MH and MUH phenotypes may present different categories of body mass index (BMI), as in the case of a subject with MUH normal weight and MH obesity. The factors associated with the development of the MUH profile still need further clarification; however, it is assumed that the distribution of body fat and lifestyle habits, such as physical activity, smoking and alcohol consumption, may contribute to this outcome. [4,5].

Regarding the practice of physical activity, it has been observed that different modalities, intensities and volumes of training can result in cardiometabolic improvement and prevention of the MUH phenotype [6,7]. Martinéz-Goméz et al. [8], in a cohort study with individuals of both sexes and different BMI classes, identified an inverse association between the practice of leisure-time physical activity and the incidence of the MUH phenotype among those with MH normal weight and overweight. In addition, it was observed that the increase in the level of physical activity contributed to the transition from the MUH to MH phenotype over a period of six years. Other studies [9,10] have shown that higher levels of cardiorespiratory fitness are associated with the MH phenotype, regardless of BMI. Despite these findings, to date, there are no studies investigating the relationship between resistance training (RT) practice and MUH and MH phenotypes.

In menopausal women, the practice of RT is recommended, as it promotes the reduction in body adiposity and can promote the increase in fat-free mass, especially musculoskeletal mass, which, as well as lipid metabolism, are affected by the reduction synthesis of female sex hormones [11,12]. In this sense, Nunes et al. [13], in a 16-week longitudinal study with 32 menopausal women, demonstrated a reduction in the concentrations of glycated hemoglobin, total cholesterol (TC) and low-density lipoprotein cholesterol (LDL-c) among those who were submitted to the practice of RT, when compared to those who performed another type of exercise. Similarly, Son et al. [14] identified a reduction in BMI, total body mass and percentage of body fat (%BF) in menopausal women submitted to 12 weeks of RT, when compared to those who did not perform any type of exercise during the study period.

The World Health Organization [1] recommends the practice of RT for at least two days a week for the adult and elderly population, without specifications for menopausal women, and highlights that there is a dose–response relationship between the practice of physical activity and cardiometabolic outcomes. However, considering that the end of reproductive life can interfere with anabolic capacity due to hormonal changes and, consequently, alterations in the metabolic profile of these women, identifying physical exercise variables, including weekly frequency and duration, is important to enhance the benefits related to RT practice [1,15].

Thus, considering that the body and metabolic changes that occur after menopause are risk factors for the MUH phenotype and that there are no studies evaluating the relationship between these components and RT in this population, the objective of the present study is to evaluate the dose–response relationship of RT with lipid profile, body composition and metabolic phenotypes in menopausal women.

## 2. Materials and Methods

### 2.1. Study Type and Sample Characteristics

An observational and cross-sectional study was carried out with a convenience sample, consisting of women practicing resistance exercises during menopause in a gym located in the city of Belo Horizonte, Minas Gerais, Brazil. The following inclusion criteria were used: (a) an age group between 45 and 65 years; (b) a clinical diagnosis of menopause (absence of menstruation for a period ≥ 12 months); and (c) the practice of resistance exercises for at least six months and twice a week. The following exclusion criteria were adopted: (a) limitation or physical disability that prevented the measurement of anthropometric and body composition data; (b) previous bariatric surgery; (c) a history of cancer or thyroid diseases; (d) undergoing sex and/or thyroid hormone replacement; (e) smoking or having cigarettes in the last six months; (f) consumption of alcoholic beverages in excess (one dose of alcohol/day, equivalent to 15 g of alcohol/day); and (g) practice of other types of physical exercise, in addition to resistance exercises.

The study’s protocol complied with the Declaration of Helsinki for conducting medical research involving human subjects and was approved by the Research Ethics Committee of Salgado de Oliveira University, located in the main unit of the Institution, in the city of Niterói, Rio de Janeiro, Brazil (protocol 99/2010). All participants were informed about the research objectives and procedures and signed the consent form.

### 2.2. Data Collection and Analyzed Variables

Data collection was carried out between March and April 2019 by a trained multidisciplinary team (doctors, nurses and physical education professionals who took part in the study) and in a previously agreed laboratory. Invitations to participate in the study were made available on the academy’s bulletin boards. Following that, information was obtained from possible participants through a registration form. After this stage, those who met the inclusion criteria answered the questionnaire prepared for the research and were referred for clinical, laboratory and body composition exams.

#### 2.2.1. Blood Pressure (BP)

BP was evaluated following the recommendations of the Brazilian Guidelines on Arterial Hypertension [16] using a UNITEC mercury column manometer, with cotton cuff and Velcro closure. At the time of measurement, the sample members had an empty bladder, had not ingested alcoholic beverages, coffee or food and had not exercised for at least one hour. Two visits to the doctor’s office were carried out and in each of them three measurements were performed, with an interval of one to two minutes between them, in a sitting position. The BP value considered was equivalent to the average of the last two measurements. The diagnosis of arterial hypertension was made by a physician who was part of the research team. The diagnosis of systemic arterial was reached when systolic blood pressure (SBP) ≥ 140 mmHg and diastolic blood pressure (DBP) ≥ 90 mmHg were obtained.

#### 2.2.2. Lipid Profile and Fasting Blood Glucose

Serum concentrations of TG, HDL-c, LDL-c, total cholesterol and blood glucose were obtained by venipuncture after a 12 h fast. Blood samples were analyzed by the enzymatic calorimetric method (Gold Analyze^®^, Belo Horizonte, Minas Gerais, Brazil). Lipid variables and blood glucose were classified according to the Sociedade Brasileira de Cardiologia [17] and the Sociedade Brasileira de Diabetes [18], respectively (SBD, 2020). The following inadequacy cut-off points were considered: TG ≥ 150 mg/dL; HDL-c ≤ 50 mg/dL; LDL-c ≥ 160 mg/dL; TC ≥ 1 90 mg/dL; and blood glucose ≥ 126 mg/dL.

#### 2.2.3. Anthropometric and Body Composition Assessment

The assessment of body composition was performed using the gold standard method of dual energy X-ray absorptiometry (DEXA), brand GE^®^ lunar I, model 40782 (Madison, WI, USA), with participants wearing tops and shorts. This method evaluates the body components from x-ray beams that cross the body in the posteroanterior direction and subdivides it into specific lines, providing, in an accurate way, information about bone mineral density, amounts of fat and fat free masses and body fat distribution [19,20].

Based on the results obtained by DEXA, the following variables were considered: total body mass, fat mass (FM) in kilograms, body fat percentage (%BF), fat free mass (FFM), trunk fat mass (TFM) in kilograms, trunk fat percentage (%TF), android fat percentage (%AF), gynoid fat percentage (%GF) and android/gynoid ratio (A/G). Among the components analyzed, the %BF was classified according to the cutoff point established by the Association of Clinical Endocrinologists and the American College of Endocrinology (AACE/ACE), that consider the inadequacy cutoff of ≥35% [21]. Body composition and height variables (in meters—m) were calculated using the Encore 2007 program, version 11.3.(Madison, WI, USA)

Additionally, the body mass index (BMI) was calculated by dividing the body mass (in kilograms—kg) by height (in square meters—m^2^). The classification of this variable was performed considering the cutoff points proposed by the WHO [22]: low weight, <18.5 kg/m^2^; eutrophy, between 18.5 and 24.9 kg/m^2^; overweight, between 25.0 and 29.9 kg/m^2^; grade I obesity, between 30.0 and 34.9 kg/m^2^; grade II obesity, between 35.0 and 39.9 kg/m^2^; grade III obesity, ≥40.0 kg/m^2^. For statistical purposes, the following categories were considered: low weight < 18.5 kg/m^2^; eutrophy, between 18.5 and 24.9 kg/m^2^; and overweight ≥ 25.0 kg/m^2^.

The assessment of abdominal adiposity was based on the waist circumference (WC), body shape index (BSI) and conicity index (CI), while visceral adiposity was estimated by visceral adiposity index (VAI) and lipid accumulation product (LAP).

The measurement of WC was performed at the end of a deep inspiration, using an inextensible measuring tape (precision of 0.1 cm), positioned at the midpoint between the lower costal margin and the iliac crest, as recommended by the I Brazilian Guidelines for Diagnosis and Treatment of Metabolic Syndrome [23]. The value > 88 cm was considered as the cut-off point for WC inadequacy, as suggested by the NCEP-ATP III [24]. The BSI was calculated using the formula proposed by Krakauer and Krakauer [25]: BSI = WC/BMI^2/3^ × height^1/2^; and the CI as recommended by Valdez [26]: CI = WC/0.109 × √body mass/height.

The VAI calculations proposed by Amato et al. [27] were utilized, using the variables WC (in centimeters—cm), BMI and serum concentrations of TG and HDL-c (in mmol/L), as demonstrated by the formula: $\mathrm{VAIwomen}=\left(\frac{\mathrm{WC}}{36.58+\left(1.89\times \mathrm{BMI}\right)}\right)$×$\left(\frac{\mathrm{TG}}{0.81}\right)$×$\left(\frac{1.52}{\mathrm{HDL}-c}\right)$. LAP was calculated considering WC (cm), TG (in mmol/L) and a constant of 58, referring to a minimum WC value, which comprises only the number of abdominal muscles, viscera and bone content: LAP women = (WC−58) × (TG) [28]. VAI and LAP were classified according to the cut-off point established by Eickemberg et al. [29] for the Brazilian population: inadequacy of VAI: ≥ 1.44; inadequacy of LAP: ≥ 22.64.

#### 2.2.4. Practice of Resistance Training (RT)

Muscular resistance training occurred independently of the present research, following recommendations from the American College of Sports Medicine, for the adult population, which establishes that exercises with free weights and single or multi-joint equipment, with an intensity of 70–100% of a maximum repetition and variations of repetitions and series, depending on the level of practice of the subjects [30]. Specific characteristics related to the volume of RT were assessed using a previously designed questionnaire, which considered frequency and time (per session, weekly and years), as performed by Burrup et al. [9] and Brellenthin et al. [31]. The sample was divided into: <2 years and ≥2 years; <300 min per week and ≥300 min per week; and ≤3 days per week and >3 days per week.

#### 2.2.5. Metabolic Phenotypes

The MH and MUH phenotypes were classified according to the Comorbidity Criteria [32], in which individuals who do not have type 2 diabetes mellitus, systemic arterial hypertension or dyslipidemia are considered MH. To this end, the following considerations were made: (a) for the diagnosis of systemic arterial hypertension: self-reported medical diagnosis; use of antihypertensives; or SBP: ≥140 mmHg; DBP: ≥90 mmHg during the assessment [16]; (b) diagnosis of type 2 diabetes mellitus: fasting blood glucose concentrations ≥ 126 mg/dL, or blood glucose after two hours of consumption of 75 g of anhydrous glucose ≥ 200 mg/dL, or glycated hemoglobin > 6.5%, or oral/subcutaneous use of hypoglycemic medication, or self-report of previous diagnosis of the disease [18]; (c) diagnosis of dyslipidemia: HDL-c: <50 mg/dL; TG: ≥150 mg/dL, or use of medication for lipid control [17].

### 2.3. Statistical Analysis

A sample size calculation was performed, based on data from a previous pilot study carried out with 10 women in a simple random sample, to estimate the population standard deviation. After the pilot study, the sample size was defined using the formula: *n* ≥ ((Zα/2*σ)/d)^2^, where: Zα/2 = Z value (for a confidence level of 95 %, Z = 1.96); *n* = number of subjects; σ = estimated variance; d = maximum estimation error. A sample consisting of 31 women was obtained as the ideal minimum size [33].

For data analysis, the normality of continuous variables was verified using the Shapiro–Wilk test, from which the use of parametric tests was chosen. The characterization of the variables in the total sample was demonstrated by the mean and standard deviation. Comparison of continuous variables according to metabolic phenotype categories (MH; MUH), time of RT practice in years (<2 years; ≥2 years), weekly time (<300 min/week; ≥300 min per week) and weekly frequency (<3 days/week; ≥3 days/week) was performed using the Student’s *t*-test. The association between categorical variables was analyzed by the chi-square test (x^2^) or Fisher’s exact test, when any of the groups presented low frequency (*n* ≤ 5), and by the prevalence ratio. The correlation between continuous variables was analyzed by Pearson’s Correlation and the correlation factor (r) was classified as strong (for values above 0.6) moderate (for values between 0.4 and 0.59) or low (for values below 0.39). A significance level of 5% was adopted (*p*-value ≤ 0.05) and the analyzes were performed using the SPSS program, version 21 (Armonk, NY, USA).

## 3. Results

### 3.1. Characteristics of the Total Sample and According to MH and MUH Phenotypes

A total of 189 women volunteered for the study, of which 76 were considered eligible. Of these, 51 underwent laboratory analysis and 44 underwent body composition assessment. Thirty-one women performed both exams, which is the sample number of the present study.

In the total sample, the mean age was 52.29 ± 4.56 years, BMI was 26.14 ± 4.93 kg/m^2^, time in years of RT was 11.29 ± 14.11, the weekly RT time was 289.35 ± 216.22 min and RT weekly frequency was 4.00 ± 1.15 days. Among the clinical and metabolic variables analyzed, MH women had a higher mean HDL-c when compared to MUH ones (MH: 72.13 ± 13.59 mg/dL; MUH: 48.37 ± 7.55 mg/dL, *p* < 0.01) (Table 1).

Regarding the distribution of body fat, MUH women had higher means of VAI (MH: 0.81 ± 0.37; MUH: 1.63 ± 0.85, *p* = 0.03) and TFM (MH: 11.33 ± 5.48 kg; MUH: 17.32 ± 8.08 kg, *p* = 0.02), A/G (MH: 0.79 ± 0.15; MUH: 0.98 ± 0.22, *p* = 0.01), and lower %BF means (MH: 50.46 ± 5.14; MUH: 41.80 ± 15.02, *p* = 0.02) when compared to MH women (Table 1). There was also a 75.0% frequency of excess weight, represented by BMI, 25.0% inadequacy of WC and 100% inadequacy of %BF among MUH women. In MH women, the prevalence of those inadequacies was of 47.8%, 13.0% and 78.3%, respectively (data not shown).

### 3.2. Comparison between Anthropometric Indices and Body Composition According Differences of Volume RT

Women with RT practice equal to or greater than two years had lower TFM than those with fewer years of practice of these exercises (<2 years: 15.33 ± 7.56 kg; ≥2 years: 10.57 ± 4.87, *p* = 0.04); the same was observed when considering the weekly frequency (≤3 days: 16.31 ± 7.46; >3 days: 10.98 ± 5.49, *p* = 0.03). For those with a higher weekly frequency, there was also a lower mean of %BF when compared to women with less practice in days/week (≤3 days: 41.40 ± 6.10; >3 days: 36.06 ± 5.46, *p* = 0.01) (Table 2).

Regarding FFM and MM, those with more than two years of RT practice had lower means in these components when compared to women with less time of practice (FFM: ≤2 years: 42.71 ± 5.93 kg, >2 years: 41.21 ± 4.83 kg, *p* = 0.44; MM: ≤2 years: 40.36 ± 5.70 kg, >2 years: 38.91 ± 4.66 kg, *p* = 0.44). When considering the time in minutes per week, there were lower FFM and MM means among those with practice equal to or greater than 300 min/week when compared to those with less weekly RT time (FFM: <300 min: 42.65 ± 5.53 kg; ≥300 min: 39.87 ± 4.49 kg, *p* = 0.21; MM: <300 min: 40.30 ± 5.35 kg, ≥300 min: 37.64 ± 4.23 kg, *p* = 0.21). The inverse was observed when analyzing the weekly frequency (FFM: ≤3 days: 41.43 ± 6.10 kg, >3 days: 42.21 ± 5.05 kg, *p* = 0.70; MM: ≤3 days: 39.06 ± 5.97 kg, >3 days: 39.91 ± 4.79 kg, *p* = 0.66) (Table 2).

Regarding visceral adiposity, women with a shorter time in years and weekly frequency of RT had higher means of VAI and LAP when compared to their respective counterparts (VAI: <2 years: 1.06 ± 0.57, ≥2 years: 0.98 ± 0.70, *p* = 0.74; ≤3 days/week: 1.14 ± 0.76, <3 days: 0.96 ± 0.56, *p* = 0.45; LAP: <2 years: 24.14 ± 17.59; ≥2 years: 19.30 ± 12.84, *p* = 0.38; ≤3 days/week: 24.84 ± 14.93, >3 days/week: 19.89 ± 15.53, *p* = 0.39) (Table 2).

### 3.3. Frequencies of Adequacy and Inadequacy Acoording Differences of Volumes RT

A higher frequency of inadequacy in these variables in women with less time in years (VAI: 20.0%, LAP: 33.3%), weekly time (VAI: 17.4%, LAP: 30.4%) and weekly frequency of RE (VAI: 18.2%, LAP: 45.5%) when compared to their respective categories was also observed (≥2 years = VAI: 12.5%, LAP: 25.0%; ≥300 min/week: VAI: 12.5%, LAP: 25.0%; ≤3 days/week = VAI: 15.0%, LAP: 20.0%), although with no statistically significant difference. The same was observed for WC, %BF and BMI. It is noteworthy, however, that there was also a high inadequacy of %BF in 75.0%, 75.0% and 80.0% of women with RT practice greater than or equal to two years, greater than or equal to 300 min per week and greater than three days per week, respectively (Figure 1).

### 3.4. Association between Lipid Profile and Metabolic Phenotype According Differences of Volume RT

Among women who practiced RT for less than two years (48.4%), there was a greater inadequacy of the analyzed lipid variables, except TC (<2 years: 53.3%, ≥2 years: 62.5%, *p* = 0.60), and higher frequency of the MUH phenotype (<2 years: 33.3%, ≥2 years: 18.7%, *p* = 0.43). Regarding the weekly time of RT practice, 74.2% of the women practiced less than 300 min/week. Among these, there was a higher frequency of inadequacy of TG (<300 min: 8.3%; ≥300 min: 0.0%, *p* = 1.00) and other lipid variables, except HDL-c (<300 min: 17.4%, ≥300 min: 25.0%, *p* = 0.63), which was higher in the group with weekly practice equal to or greater than 300 min. The frequency of the MUH phenotype among women with a shorter weekly RT time was 26.1% versus 25.0% of those with a weekly RT time greater than or equal to 300 min. Regarding the weekly frequency, 64.5% performed RT more than three times a week. Among these, there was less inadequacy of lipid profile variables and MUH phenotype (≤3 days/week: 36.4%, >3 days/week: 20.0%, *p* = 0.40) than among those with a lower weekly frequency (Table 3).

### 3.5. Correlations and Prevalence Ratio between Body and Lipid Variables According Differences of Volume RT

There was a moderate, negative and significant correlation between time in years with FM, FFM, %BF and A/G; and between weekly frequency of RT with FM and %BF. A weak, negative and significant correlation between time in years and BMI, %TF and %GF, in addition to a weak, positive and significant correlation between this and HDL-c was also identified (Table 4).

Data on the prevalence ratio between metabolic phenotypes, lipid profile and body variables with time in years, weeks and weekly frequency of RT are shown in Table 5. It was identified that women with less time in years and minutes per week of RT have a prevalence of being overweight by a BMI 2.26 (CI 95%: 0.92–5.56) and 1.54 (CI 95%: 0.95–2.51) times higher than those with longer time in years and weeks, respectively, of these exercises. Regarding the weekly frequency, those with a weekly frequency equal to or less than three days of RT have a prevalence of having an inadequacy of %BF 3.25 (CI 95%: 0.83–12.61) times higher than those with a weekly frequency of more than three days of practice of RT. Lower prevalence ratios for the MUH phenotype and inadequacies of the lipid profile were observed in the three categories of practice of these exercises, when compared to body variables.

## 4. Discussion

The present study is unprecedented in that it is the first to assess the dose–response relationship of RT with lipid profile, body composition and metabolic phenotypes during menopause. Among the main findings, it is noteworthy that greater time in years, weekly time and weekly frequency of RT are associated with lower body adiposity, and that there is a dose–response relationship of RT with lipid profile and the MUH phenotype in women of non-reproductive age.

It is well established that the reduction in the synthesis of female sex hormones by ovarian follicular cells during menopause promotes physiological and body composition changes that predispose to the development of comorbidities and the MUH phenotype [2,34]. In the present study, this metabolic risk profile was prevalent in a smaller proportion of the sample; however, given the lack of standardization of classification criteria, the percentages of MH and MUH phenotypes between studies can be varied. It is known, however, that some factors, such as physical inactivity and accumulation of abdominal fat, are associated with the MUH phenotype [35].

Data from this research corroborate the evidence that MUH individuals have greater body adiposity, though the frequency of excess weight (by BMI) and the inadequacies of WC and %BF among MH women should be highlighted, given the transitory aspect of this phenotype [36]. WC is a widely used anthropometric measure to assess body fat distribution and cardiometabolic risk. However, as it is influenced by body mass and height, some authors recommend that these components be evaluated by other parameters [37,38].

In the present study, the distribution of body fat was evaluated using methods that are easily applicable in professional practice, such as anthropometry or abdominal and visceral adiposity indices, as well as the gold standard DEXA method. In all parameters, except CI, BSI and %GF, higher averages were identified among MUH women. Corroborating these findings, Lwow et al. [4], in a sample made up of menopausal women and using four different methods of classification of metabolic phenotypes, found higher means of WC, AFM, GFM, A/G and LAP among MUH women.

In the context of studies on metabolic phenotypes and menopause, there is no information on the association between CI, BSI and the MUH profile. However, Rasaei et al. [39], in a study with a sample of women of reproductive age, identified a significant relationship between BSI and the MUH phenotype (area under the curve: 0.60, *p* < 0.05). Regarding CI, in a study with menopausal women, Gadelha et al. [40] found that this index has a significant correlation with metabolic syndrome (area under the curve: 0.74, *p* < 0.01). It is noteworthy, however, that the findings on the relationship between the aforementioned indices and cardiometabolic risk are still scarce and need further investigation [41,42].

The relationship between metabolic phenotypes and physical activity has been studied by several authors [8,9]; however, there is still no evidence about these and the practice of RT. In the current study, a dose–response relationship was identified between this type of physical exercise and the metabolic risk profile. Similarly, when comparing the effects of different frequencies of resistance training on cardiometabolic risk in overweight women after 24 weeks of intervention, Campa et al. [43] observed that those with a weekly frequency equal to or greater than three days showed better lipid profile, lower fasting glucose, HOMA-IR, glycated hemoglobin and insulin concentrations, in addition to lower BMI, WC and %BF, when compared to those who performed RT only once a week.

Findings on the impact of RT on the lipid profile in menopausal women show conflicting results [13,44,45]. In the present study, it was possible to identify a dose–response relationship between the weekly frequency of RT and lipid variables, as seen in the lower inadequacy of these components in women who performed RT for more than three days a week. These data contradict the findings of Orsatti et al. [46], who, in a study with menopausal women undergoing a 16-week RT intervention, did not identify changes in the lipid profile when comparing different weekly frequencies.

The benefits associated with RT practice under cardiometabolic parameters can be explained by the improvement in GLUT-4 protein signaling and an increase in mitochondrial content, which contribute to greater glucose uptake by musculoskeletal tissue and fatty acid oxidation [47,48]. Additional mechanisms include increased resting energy expenditure, which, in turn, may be associated with increased post-exercise energy consumption and lean body mass [47,48]. There are numerous endocrine-metabolic pathways related to the increase in FFM, specifically MM, in response to mechanical stimulation of RT, among which it is possible to highlight the greater synthesis of anabolic hormones associated with the repair of musculoskeletal tissue, besides the greater gene expression related to muscle protein synthesis [11,49].

In the current sample, women with more time in years and minutes per week of RT had higher muscle content measured by DEXA, though the same was not observed in relation to the weekly frequency of this modality. Unlike these findings, when evaluating the body composition of adult women practicing RT, Burrup et al. [12] found that longer time in years, minutes per session and week and weekly frequency are associated with higher FFM in this population. It is noteworthy, however, that the differences found between both studies may be related to the specific characteristics of the sample, since menopausal women were analyzed in the present study, while Burrup et al. [12] evaluated women at different stages of their reproductive life.

The dose–response relationship associated with the increase in FFM from the practice of RT in menopausal women is still not well understood. A meta-analysis with the objective to evaluate the effects of this type of exercise on FFM in this population showed no difference in the gain of this component related to age, weekly frequency and intervention period of the analyzed studies [50]. Similarly, Kneffel et al. [15] identified a relationship between weekly frequency and muscle strength, but not with musculoskeletal hypertrophy, in a meta-regression on the effects of RT training frequency over these factors in older adults of both sexes. However, further studies are needed to analyze this relationship, since the depletion of estrogen concentrations, associated with ovarian follicular cell failure, significantly affects FFM, muscle strength, increases FM and makes this group more susceptible to sarcopenia and cardiometabolic diseases [43,51].

As observed in the present study, time in years and weekly frequency of RT are inversely associated with body adiposity, and there is a higher prevalence ratio for overweight (by BMI) and inadequacy of %BF among women with less time in years and weekly frequency of RT, respectively. Corroborating these findings, Campa et al. [43] demonstrate a dose–response relationship of the weekly frequency of RT with body adiposity, with a reduction in WC and %BF by 9.0% and 9.8% for menopausal women who perform RT three days a week versus 4.7% and 5.3%, respectively, among those with a lower weekly frequency. Similarly, Carvalho et al. [52] emphasize that a reduction in adipose components in menopausal women can be observed six months after the start of RT practice, thus showing that there also is a dose–response relationship of this modality with time in months.

These data are important since body adiposity is a potent risk factor for the development of cardiometabolic complications, such as DM2, SAH and atherosclerosis, among others. It is noteworthy, however, that the distribution of body adiposity in females occurs differently, according to the biological moment: during reproductive life, there is a gynoid pattern, with accumulation of fat in the peripheral region of the body, while during menopause there is a prevalence of the android profile with greater fat deposits in the central region. In this specific period, in addition to the reduction in estrogen concentrations, there is an increase in the bioavailability of testosterone. Thus, this hormonal imbalance, the presence of androgen receptors in visceral adipose tissue and changes in the activity of lipoprotein lipase and antilipolytic α-2 adrenergic receptors favor the accumulation of fat in the central region and promote changes in lipid metabolism [34,46,53,54].

Given the absence of studies that analyze the dose–response relationship of RT with body composition, lipid profile and metabolic phenotypes in menopausal women, the novelty of the present study is its main strength. In addition, extensive anthropometric and body composition research is highlighted, based on different assessment methods. However, the present study has some limitations, such as the cross-sectional model and the sample size, though relevant statistical analysis were used to minimize this characteristic. Despite this, it is noteworthy that our results may not be applicable to other populations and is recommended to carry out additional longitudinal and intervention studies to assess the dose–response relationship between RT, lipid profile, body composition and metabolic phenotypes in menopausal women, as there are still no studies on this topic in the scientific literature.

## 5. Conclusions

Time greater than two years and weekly frequency equal to or greater than three days of RT are associated with lower body adiposity in menopausal women, the first also being associated with HDL-c concentrations in this population. Such findings may contribute to the development of training protocols and provide support for clinical practice aimed at mitigating cardiometabolic and bodily damage associated with the end of reproductive life. Additional studies are needed to investigate the dose–response relationship of RT under lipid profile and metabolic phenotypes in menopausal women.

## Figures and Tables

**Figure 1 ijerph-19-10369-f001:**
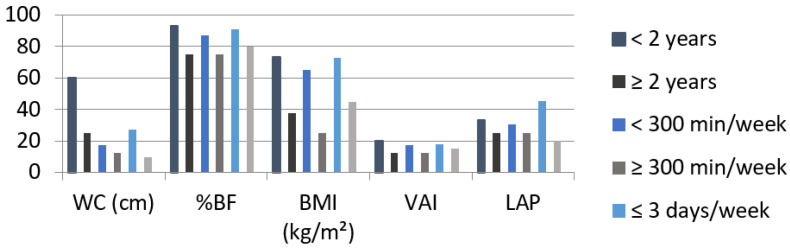
Frequency of inadequacy of anthropometric indices among women with resistance training time < 2 years, ≥2 years; <300 min per week, ≥300 min per week; and weekly frequency ≤ 3 days per week and >3 days per week. WC: waist circumference; %BF: body fat percentage; BMI: body mass index; VAI: visceral adiposity index; LAP: lipid accumulation product.

**Table 1 ijerph-19-10369-t001:** Characteristics of the total sample and according to MH and MUH phenotypes.

	Total	MH	MUH
	100% (*n* = 31)	74.2% (*n* = 23)	25.8% (*n* = 8)
	Mean ± SD	Mean ± SD	Mean ± SD
**Descriptive characteristics**			
Age (years)	52.29 ± 4.56	**51.26 ± 4.44**	**55. 25 ± 3.73 ***
BMI (kg/m^2^)	26.14 ± 4.93	25.74 ± 4.93	27.29 ± 5.09
WC (cm)	80.31 ± 7.62	79.85 ± 7.56	81.62 ± 8.14
**Resistance training**			
Time_years RT (years)	11.29 ± 14.11	**11.70 ± 13.60**	**3.81 ± 3.66 ***
Time_week RT(min/week)	289.35 ± 216.22	304.78 ± 234.20	245.00 ± 157.93
Freq_week RT (days/week)	4.00 ± 1.15	4.08 ± 1.23	3.75 ± 0.88
**Metabolic and clinic profile**			
HDL-c (mg/dL)	66.66 ± 16.14	**72.13 ± 13.59**	**48.37 ± 7.55 ***
TG (mg/dL)	82.90 ± 47.05	73.65 ± 29.88	109.50 ± 74.78
LDL-c (mg/dL)	115.00 ± 29.32	110.04 ± 26.28	129.25 ± 34.66
VLDL-c (mg/dL)	20.29 ± 11.49	18.13 ± 9.01	26.50 ± 15.84
TC (mg/dL)	201.25 ± 35.34	200.30 ± 31.30	204.00 ± 47.56
Blood glucose (mg/dL)	88.41 ± 6.21	88.69 ± 6.01	87.62 ± 7.13
SBP (mmHg)	122.70 ± 7.80	121.91 ± 8.02	125.00 ± 7.11
DBP (mmHg)	78.61 ± 5.85	79.00 ± 5.70	77.50 ± 6.54
**Anthropometric indices**			
VAI	1.02 ± 0.63	**0.81 ± 0.37**	**1.63 ± 0.85 ***
CI	1.14 ± 0.02	1.14 ± 0.02	1.13 ± 0.03
BSI	0.11 ± 0.00	0.11 ± 0.00	0.11 ± 0.00
LAP	21.64 ± 15.26	19.28 ± 13.83	28.43 ± 18.06
**Body composition**			
FM (kg)	25.06 ± 8.41	24.43 ± 8.67	26.89 ± 7.86
FFM (kg)	41.93 ± 5.35	41.68 ± 5.56	42.65 ± 4.99
MM (kg)	39.61 ± 5.15	39.36 ± 5.38	40.34 ± 4.69
%BF	37.92 ± 6.18	37.44 ± 6.68	39.42 ± 4.40
TFM (kg)	12.87 ± 6.66	**11.33 ± 5.48**	**17.32 ± 8.08 ***
%TF	36.49 ± 10.40	35.88 ± 9.27	38.25 ± 13.75
AFM (kg)	3.14 ± 5.47	1.97 ± 1.17	6.52 ± 10.32
%AF	40.87 ± 12.65	40.30 ± 10.63	42.53 ± 18.06
GFM (kg)	6.95 ± 8.81	5.53 ± 1.42	11.06 ± 17.33
%GF	48.23 ± 9.82	**50.46 ± 5.14**	**41.80 ± 15.02 ***
A/G	0.84 ± 0.19	**0.79 ± 0.15**	**0.98 ± 0.22 ***

* (*p* < 0.05) related to the comparison of MH and MUH. Data shown as mean ± standard deviation (Mean ± SD). Student’s *t*-test. MH: Metabolically healthy phenotype; MUH: Metabolically unhealthy phenotype. BMI: body mass index; WC: waist circumference; Time_years RT: time of resistance training (years); Time_week RT: weekly time of resistance training (min/week); Freq_week RT: weekly frequency resistance training (days/week); HDL-c: high-density lipoprotein cholesterol; TG: triglycerides; LDL-c: low-density lipoprotein cholesterol; VLDL-c: very low density lipoprotein cholesterol; TC: total cholesterol; SBP: systolic blood pressure; DBP: diastolic blood pressure; VAI: visceral adiposity index; CI: conicity index; BSI: body shape index; LAP: lipid accumulation product; FM: fat mass; FFM: fat free mass; MM: muscle mass; %BF: body fat percentage; TFM: trunk fat mass; %TF: percentage of trunk fat mass; AFM: android fat mass; %AF: percentage of android fat; GFM: gynoid fat mass; %GF: percentage of gynoid fat; A/G: android/gynoid ratio.

**Table 2 ijerph-19-10369-t002:** Comparison between anthropometric indices and body composition according to time in years, time in minutes per week and weekly frequency of resistance exercises.

	<2 Years	≥2 Years	<300 min/week	≥300 min/week	≤3 days/week	>3 days/week
	**48.4% (*n* = 15)**	51.6% (*n* = 16)	74.2% (*n* = 23)	25.8% (*n* = 8)	35.5% (*n* = 11)	64.5% (*n* = 20)
	Mean ± SD	Mean ± SD	Mean ± SD	Mean ± SD	Mean ± SD	Mean ± SD
**Anthropometric indices**						
BMI (kg/m^2^)	27.41 ± 5.37	24.95 ± 4.32	26.77 ± 5.37	24.32 ± 2.96	27.49 ± 5.99	25.40 ± 4.23
WC (cm)	82.06 ± 8.65	78.67 ± 6.35	81.17 ± 8.33	77.84 ± 4.62	82.31 ± 9.64	79.21 ± 6.26
VAI	1.06 ± 0.57	0.98 ± 0.70	0.98 ± 0.69	1.13 ± 0.44	1.14 ± 0.76	0.96 ± 0.56
LAP	24.14 ± 17.59	19.30 ± 12.84	22.24 ± 16.93	19.94 ± 9.66	24.84 ± 14.93	19.89 ± 15.53
CI	1.13 ± 0.02	1.14 ± 0.03	1.13 ± 0.03	1.14 ± 0.02	1.14 ± 0.03	1.14 ± 0.02
BSI	0.11 ± 0.00	0.11 ± 0.00	0.11 ± 0.00	0.11 ± 0.00	0.11 ± 0.00	0.11 ± 0.00
**Body composition**						
FM (kg)	**27.71 ± 8.93**	**22.58 ± 7.32 ***	26.13 ± 8.86	22.00 ± 6.51	28.49 ± 9.11	23.18 ± 7.59
FFM (kg)	42.71 ± 5.93	41.21 ± 4.83	42.65 ± 5.53	39.87 ± 4.49	41.43 ± 6.10	42.21 ± 5.05
MM (kg)	40.36 ± 5.70	38.91 ± 4.66	40.30 ± 5.35	37.64 ± 4.23	39.06 ± 5.97	39.91 ± 4.79
%BF	39.96 ± 5.44	36.08 ± 6.37	38.54 ± 6.46	36.27 ± 5.23	**41.40 ± 6.10**	**36.06 ± 5.46 ^‡^**
TFM (kg)	**15.33 ± 7.56**	**10.57 ± 4.87 ***	13.61 ± 7.10	10.76 ± 4.96	**16.31 ± 7.46**	**10.98 ± 5.49 ^‡^**
%TF	38.43 ± 11.58	34.68 ± 9.17	36.80 ± 11.06	35.62 ± 8.86	39.74 ± 12.97	34.71 ± 8.54
AFM (kg)	**4.51 ± 7.68**	**1.86 ± 1.12 ***	3.60 ± 6.30	1.82 ± 0.98	5.39 ± 8.86	1.91 ± 1.18
%AF	43.31 ± 14.91	38.60 ± 10.05	41.29 ± 13.64	39.67 ± 9.91	44.24 ± 16.42	39.03 ± 10.03
GFM (kg)	8.97 ± 12.51	5.06 ± 1.08	7.68 ± 10.17	4.86 ± 0.76	10.15 ± 14.59	5.19 ± 1.14
%GF	47.98 ± 12.65	48.46 ± 4.92	47.99 ± 10.56	48.91 ± 4.60	47.19 ± 14.65	48.80 ± 4.82
A/G	0.88 ± 0.19	0.79 ± 0.18	0.85 ± 0.18	0.81 ± 0.21	0.91 ± 0.17	0.80 ± 0.18

* (*p* < 0.05) related to the comparison of <2 years and ≥2 years; **^‡^** (*p* < 0.05) related to the comparison of ≤3 days/week e > 3 days/week. Data show as mean ± standard deviation (Mean ± SD). Student’s *t*-test. BMI: body mass index; WC: waist circumference; VAI: visceral adiposity index; LAP: lipid accumulation product; CI: conicity index; BSI: body shape index; FM: fat mass; FFM: fat free mass; MM: muscle mass; %BF: body fat percentage; TFM: trunk fat mass; %TF: percentage of trunk fat mass; AFM: android fat mass; %AF: percentage of android fat; GFM: gynoid fat mass; %GF: percentage of gynoid fat; A/G: android/gynoid ratio.

**Table 3 ijerph-19-10369-t003:** Association between lipid profile and metabolic phenotypes according to time in years, in minutes per week and weekly frequency of RT.

	<2 Years	≥2 Years	<300 min/week	≥300 min/week	≤3 days/week	>3 days/week
	48.4% (*n* = 15)	51.6% (*n* = 16)	74.2% (*n* = 23)	25.8% (*n* = 8)	35.5% (*n* = 11)	64.5% (*n* = 20)
**Lipid profile**						
HDL-c (mg/dL)						
Adequate	73.3 (11)	87.5 (14)	82.6 (19)	75.0 (6)	72.7 (8)	85.0 (17)
Inadequate	26.7 (4)	12.5 (2)	17.4 (4)	25.0 (2)	27.3 (3)	15.0 (3)
TG (mg/dL)						
Adequate	93.3 (14)	93.8 (15)	91.3 (21)	100.0 (8)	90.9 (10)	95.0 (19)
Inadequate	6.7 (1)	6.2 (1)	8.7 (2)	0.0 (0)	9.1 (1)	5.0 (1)
LDL-c (mg/dL)						
Adequate	86.7 (13)	93.8 (15)	87.0 (20)	100.0 (8)	81.8 (9)	95.0 (19)
Inadequate	13.3 (2)	6.2 (1)	13.0 (3)	0.0 (0)	18.2 (2)	5.0 (1)
TC (mg/dL)						
Adequate	46.7 (7)	37.5 (6)	34.8 (8)	62.5 (5)	36.4 (4)	45.0 (9)
Inadequate	53.3 (8)	62.5 (10)	65.2 (15)	37.5 (3)	63.6 (7)	55.0 (11)
**Metabolic phenotype**						
MH	66.7 (10)	81.3 (13)	73.9 (17)	75.0 (6)	63.6 (7)	80.0 (16)
MUH	33.3 (5)	18.7 (3)	26.1 (6)	25.0 (2)	36.4 (4)	20.0 (4)

There was no statistically significant *p*-value (*p* < 0.05) between lipid profile and metabolic phenotypes according to time in years, in weeks and weekly frequency of RT. Fisher’ exact test; HDL-c: high-density lipoprotein cholesterol; TG: triglycerides; LDL-c: low-density lipoprotein cholesterol; TC: total cholesterol; MH: metabolically healthy; MUH: metabolically unhealthy.

**Table 4 ijerph-19-10369-t004:** Correlation between body and lipid variables, time in years, in minutes per week and weekly frequency of resistance training in the total sample.

		Time_Years RT		Time_Week RT		Freq_Week RT	
		r		r		r	
**Anthropometric indices**							
BMI (kg/m^2^)		**−0.36 ***		−0.27		−0.25	
WC (cm)		−0.32		−0.23		−0.31	
CI		0.23		0.23		0.17	
BSI		0.27		0.14		0.23	
VAI		−0.20		0.03		−0.01	
LAP		−0.24		−0.09		−0.16	
**Body composition**							
FM (kg)		**−0.45 ***		−0.30		**−0.41 ***	
FFM (kg)		−0.21		−0.27		−0.20	
MM (kg)		−0.21		−0.27		−0.20	
TFM (kg)		**−0.42**		−0.27		**−0.38**	
AFM(kg)		−0.21		−0.17		−0.23	
GFM(kg)		−0.17		−0.15		−0.21	
%TF		**−0.37 ***		−0.14		−0.25	
%AF		**−0.39 ***		−0.14		−0.24	
%GF		−0.09		0.04		0.00	
%BF		**−0.49 ***		−0.25		**−0.41 ***	
A/G		**−0.44 ***		−0.19		−0.29	
**Lipid profile**							
HDL-c (mg/dL)		**0.39 ***		−0.03		−0.03	
TG (mg/dL)		−0.11		0.02		−0.04	
LDL-c (mg/dL)		−0.27		−0.21		−0.16	
VLDL-c (mg/dL)		−0.00		−0.10		−0.16	
TC (mg/dL)		−0.04		−0.24		−0.20	

Pearson’ Correlation. (*) *p* < 0.05. Time_years RT: time of resistance training (years); Time_week RT: weekly time of resistance training (minutes/week); freq_week RT: weekly frequency resistance training (days/week); r: correlation’ factor. BMI: body mass index; WC: waist circumference; CI: conicity index; BSI: body shape index; VAI: visceral adiposity index; LAP: lipid accumulation product; FM: fat mass; FFM: fat free mass; MM: muscle mass; TFM: trunk fat mass; AFM: android fat mass; GFM: gynoid fat mass; %TF: percentage of trunk fat mass; %AF: percentage of android fat; %GF: percentage of gynoid fat; %BF: body fat percentage; A/G: android/gynoid ratio; HDL-c: high-density lipoprotein cholesterol; TG: triglycerides; LDL-c: low-density lipoprotein cholesterol; VLDL-c: very low density lipoprotein cholesterol; TC: total cholesterol.

**Table 5 ijerph-19-10369-t005:** Prevalence ratio between body variables, lipid profile and metabolic phenotypes and time in years, in minutes per week per and weekly frequency of RT.

		Time_Years RT	
	PR		IC 95% (Lower–Upper)
Metabolic phenotypes	1.43		(0.70–2.92)
HDL-c (mg/dL)	1.51		(0.73–3.10)
TG (mg/dL)	1.03		(0.24–4.35)
LDL-c (mg/dL)	1.43		(0.58–3.50)
CT (mg/dL)	0.82		(0.40–1.69)
BMI (kg/m^2^)	2.26		(0.92–5.56)
WC (cm)	1.89		(1.01–3.54)
%BF	1.98		(0.81–4.86)
VAI	1.30		(0.56–2.97)
LAP	1.22		(0.58–2.56)
A/G	1.30		(0.56–2.97)
		Time_week RT	
Metabolic phenotypes	1.01		(0.63–1.62)
HDL-c (mg/dL)	0.87		(0.47–1.61)
TG (mg/dL)	1.38		(1.10–1.72)
LDL-c (mg/dL)	1.40		(1.10–1.77)
CT (mg/dL)	1.35		(0.84–2.18)
BMI (kg/m^2^)	1.54		(0.95–2.51)
WC (cm)	1.09		(0.66–1.79)
%BF	0.93		(0.62–1.41)
VAI	1.09		(0.66–1.79)
LAP	1.06		(0.69–1.64)
A/G	1.09		(0.66–1.79)
		Freq_week RT	
Metabolic phenotypes	1.64		(0.64–4.15)
HDL-c (mg/dL)	1.56		(0.58–4.17)
TG (mg/dL)	1.45		(0.33–6.33)
LDL-c (mg/dL)	2.07		(0.79–5.44)
CT (mg/dL)	1.26		(0.46–3.43)
BMI (kg/m^2^)	2.19		(0.71–6.74)
WC (cm)	1.95		(0.77–4.88)
%BF	3.25		(0.83–12.61)
VAI	1.15		(0.34–3.82)
LAP	2.03		(0.83–5.00)
A/G	1.95		(0.77–4.88)

PR: prevalence ratio. CI 95%: 95% confidence interval. Time_years RT: time of resistance training (years); time_week RT: weekly time of resistance training (minutes/week); freq_week RT: weekly frequency resistance training (days/week); HDL-c: high-density lipoprotein cholesterol; TG: triglycerides; LDL-c: low-density lipoprotein cholesterol; TC: total cholesterol; BMI: body mass index; WC: waist circumference; %BF: body fat percentage; VAI: visceral adiposity index; LAP: lipid accumulation product; A/G: android/gynoid ratio.

## Data Availability

The datasets generated during and/or analyzed during the current study are available from the corresponding author upon reasonable request.

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
