# Peer review of "Dose–Response Relationship of Resistance Training on Metabolic Phenotypes, Body Composition and Lipid Profile in Menopausal Women"

_ijerph, 2022, doi:10.3390/ijerph191610369_

Round 1

Reviewer 1 Report

Introduction: More detailed exploration of exercise and metabolic health is needed. How does resistance training compare to other training types. What sort of resistance training protocols have been used. Also need a stronger exploration of what is meant by metabolically unhealthy as this seems to be a catch all when what is predominantly being looked at in the paper is fat metabolism. A more detailed introduction to the full topic area is needed.

Methods: The collection of resistance training data seems to simplistic to me. Why was no attempt taken to record the intensity of sessions since it is well established that exercise intensity is a key variable for metabolic adaptation to exercise. Was any attempt made to determine the extent of cardiovascular work carried out in visits to the gym as this will have a major impact on metabolic health. Was any attempt made to control for job type.

Results: No presentation of blood pressure data

Overall: Unfortunately I do not see the novelty in what has been done. I do not see what new findings you are adding to the broader literature on exercise and health. Therefore I cannot recommend publication at this time.

Reviewer 2 Report

The manuscript “Dose-response Relationship of Resistance Training on Metabolic Phenotypes, Body Composition and Lipid Profile in Menopausal Women” provides information on the effects of resistance training on anthropometric and metabolic parameters in menopausal women.

The strength of the work is the high practical relevance of the threshold values identification that may result in a clinically meaningful improvement in the health of menopausal women associated with a reduction in CVD risk.

Comments, Concerns, and Suggestions:

  1. The research was conducted on a relatively small group, therefore authors performed a sample size calculation based on data from a previous pilot study to justify the choice of statistical methods. Please provide a citation for the sample size calculation.
  2. The correlation between the time of RT in years and HDL-c is moderate. Maybe the conclusion about the association of greater time of resistance training with HDL-c concentration is too far-riching when considering the relatively small size of the group?
  3. Table 3, page 8 – the phrase “Perfil lipídico” should be translated into English.

Reviewer 3 Report

The article is devoted to a topic of substantial importance and matches the range of issues generally covered by the  IJERPH.
As regards the formal aspect, the article has been subject to a well-thought-out preparation procedure and conforms to the standards ofthe  IJERPH. The title of the paper corresponds to the issues it addresses. The authors have employed a method which complies with the criteria to be met by scientific papers. Study findings have been interpreted correctly. Statistical methods were used appropriately.
After minor corrections, the article may be approved for publication. In order to increase the value of its content, I suggest the following:

 - Comment 1.  Its significance for foreign readers should be clarified, given the fact that the  IJERPH is a journal with an international audience
 - Comment 2. Please determine the exact units where the authors were given the permission to conduct their studies

Round 2

Reviewer 1 Report

I thank the authors for the changes to the paper. There is a significant improvement in the paper but I still fail to see the novelty in the findings and cannot recommend publication.

Author Response

Dear reviewer,

In this version, after the second round of contributions, we added information in the introduction in order to give greater visibility to the novelty of our work regarding the relationship between resistance training (RT) and Metabolically Healthy (MH) and Metabolically unhealthy (MUH) phenotypes in menopausal women (purple section ).

We emphasize that previously studies were carried out that related the practice of leisure-time physical activity with metabolic phenotypes in populations of both sexes and different age groups. However, in the specific population of menopausal women, these analyzes have not yet been performed, especially with regard to RT.

We consider that the relevance of this work lies 1) in the novelty related to the approach on the dose-response relationship of RT on metabolic phenotypes, body composition and lipid profile, in menopausal women, since, to our knowledge, there are still no studies on this topic in the scientific literature; 2) the broad anthropometric and body composition approach that can help health professionals to analyze cardiometabolic risk in different ways, especially in the absence of a gold standard method for assessing body composition such as DEXA; and 3) the ability to communicate to the general population about the benefits of this type of exercise for the cardiometabolic health of menopausal women.